# Effects of Vine Tea Extract on Meat Quality, Gut Microbiota and Metabolome of Wenchang Broiler

**DOI:** 10.3390/ani12131661

**Published:** 2022-06-28

**Authors:** Luli Zhou, Hui Li, Guanyu Hou, Jian Wang, Hanlin Zhou, Dingfa Wang

**Affiliations:** 1Tropical Crops Genetic Resources Institute, Chinese Academy of Tropical Agricultural Sciences, Haikou 571101, China; zhoull@catas.cn (L.Z.); guanyuhou@126.com (G.H.); 2College of Animal Science and Technology, Hainan University, Haikou 570228, China; lihui2022518@163.com (H.L.); wangjian901@163.com (J.W.); 3Zhanjiang Experimental Station, Chinese Academy of Tropical Agricultural Sciences, Zhanjiang 524013, China

**Keywords:** *Ampelopsis grossedentata*, broiler, meat quality, gut microflora, metabolome

## Abstract

**Simple Summary:**

The current work was to assess the effect of vine tea (*Ampelopsis grossedentata*) extract (AGE) addition on the meat quality, gut microbiota and metabolome of Wenchang broilers. It was found that diet addition with AGE could enhance the quality and flavor of Wenchang broiler.

**Abstract:**

This study investigates the effects of vine tea (*Ampelopsis grossedentata*) extract (AGE) on meat quality, gut microbiota and cecal content metabolites of Wenchang broilers. A total of 240 female Wenchang broilers aged 70 days were randomly allocated into four groups with five replicates of twelve broilers each. Broilers were fed a corn-soybean basal diet supplemented with AGE at 0 (T1), 0.2% (T2), 0.4% (T3) and 0.6% (T4) until 124 days of age. The whole feeding trial lasted 54 days. Results suggest that the content of total triglycerides and low-density lipoprotein cholesterol in serum of broilers are linearly reduced with dietary AGE supplementation (*p* < 0.05). The T3 and T4 groups had higher (*p* < 0.05) a* value in thigh and breast muscles than the T1 group. Additionally, the dietary supplementation of AGE decreased the shear force and drip loss of both thigh and breast muscles linearly (*p* < 0.05). Compared with the T1 group, AGE supplementation increased the levels of inosine monophosphate (IMP) significantly (*p* < 0.05) in both the thigh and breast muscles. Furthermore, an increase (*p* < 0.05) in the total unsaturated fatty acid (USFA), polyunsaturated fatty acids (PUFA) and the ratio of unsaturated fatty acids to saturated fatty acid (USFA: SFA) in both the thigh and breast muscles in the T3 group was observed. Higher abundance of *Bacteroidota* (*p* < 0.05) and lower abundance of *Firmicutes* (*p* < 0.05) were observed in the T3 group. The abundance of *Faecalibacterium* was significantly decreased (*p* < 0.05) in the T3 group compared with the T1 group. Cholesterol sulfate and p-cresol sulfate were identified as differential metabolites between the T1 and T3 groups. It suggested that 0.4% of AGE supplementation significantly downregulated the levels of p-cresol sulfate and cholesterol sulfate (*p* < 0.05) and the hepatic 3-hydroxy-3-methylglutaryl-CoA reductase (HMGR) activity compared with the control. Our present study demonstrates that dietary supplementation with AGE can improve the quality and flavor by increasing the IMP and PUFA content in the muscle of Wenchang broilers. Furthermore, dietary AGE supplementation with 0.4% can regulate the cholesterol metabolism of Wenchang broilers.

## 1. Introduction

Antibiotics used as growth promoters can not only improve the feed conversion efficiency of animals, but also help to improve the health status of livestock through preventive use [1]. However, the use of antibiotics has resulted in the emergence of microbes resistant to antibiotics which possibly has a serious impact on people’s health [2]. Moreover, antibiotics can not only disrupt the equilibrium among intestinal commensal populations, which result in deleterious effects for the host, but the long-term intake of antibiotics can also reduce the quality of poultry meat [3,4]. Therefore, the use of antibiotics in feed is increasingly restricted in more and more countries and areas. For example, use of antibiotics as feed additives, prophylactics and growth promoters in feed is banned in the European Union since 2006, and production of commercial feed containing antibiotics by feed enterprises is banned in China since February 2020 [5]. However, it has been proven that the withdrawal of antibiotics in animal production can induce animal growth problems and increase the risk of animal diseases [6]. Thus, alternatives to antibiotics to improve animal performance are urgently needed. Natural compounds produced from various plant sources seem a good substitutes to antibiotics [7,8].

*Ampelopsis grossedentata* (Hand.-Mazz.) W.T. Wang (AG), also known as vine tea, is mainly distributed in Southern China. AG has anti-bacterial and antioxidant properties and was widely used for the treatment of hepatitis, cough and hypertension [9,10,11], which is associated with the high levels of flavones in this plant [12]. Among the flavones, dihydromyricetin and myricetin have been confirmed as the main bioactive constituents in AG [13]. Previous studies have demonstrated that dihydromyricetin could protect endothelial cells from oxidative damage by inhibiting reactive oxygen species (ROS) production as well as regulate the levels of lipoprotein and blood lipid, and myricetin could reduce oxidative stress by decreasing the level of malondialdehyde (MDA) and increasing the levels of superoxide dismutase (SOD) and glutathione (GSH) in the intestinal tissues [14,15]. Meanwhile, several studies report that both dihydromyricetin and myricetin could reduce the production of inflammatory factors, such as IL-6, IL-1β and TNF-α triggered by inhibiting the NF-κB pathway [16,17,18,19]. Moreover, previous researchers have indicated that dihydromyricetin and myricetin could reduce atherosclerosis via suppression of cholesterol accumulation in macrophage foam cell [20,21].

However, how AG extract (AGE) influences meat quality, the lipid metabolism and gut health in broilers is rarely reported. In this study, we hypothesize that AGE could be used as a dietary supplement for broilers to regulate their lipid metabolism and improve gut health. The study objectives are to assess the effect of AGE addition on the meat quality, gut microbiota and metabolome of Wenchang broilers.

## 2. Materials and Methods

### 2.1. Animals, Experimental Design and Diet

This experiment was reviewed and approved by the Institutional Animal Care and Use Committee of the Chinese Academy of Tropical Agricultural Sciences (approval number CATAS-20201015-1), and the management of birds complied with the Laboratory Animal Requirements of Environment and Housing Facilities [22].

A total of 240 seventy-day-old female Wenchang broilers with an average initial body weight of 1.1 ± 0.02 kg were randomly allocated into four groups, each with five replicate cages (12 birds per cage). The feeding trial lasted 54 days. The control group was fed a basal diet (T1), while the broilers in the three AGE-treated groups were fed a basal diet supplemented with AGE at 0.2% (T2), 0.4% (T3) and 0.6% (T4). The added dose of AGE referred to the recommended dosage by the product manufacturer. The ingredients and nutrient content of the basal diet are presented in Table 1. Feed and water were offered on an ad libitum basis to the broilers. AGE was provided by Hainan Shunqin Biological Technology Co., Ltd. (Wenchang, China), which contained 41.5% (*w*/*w*) flavonoids. Flavonoids in AGE mainly included dihydromyricetin (36.8% *w*/*w*) and myricetin (0.83% *w*/*w*).

### 2.2. Sample Collection

Blood was collected on the morning of day 54. One broiler per replicate was randomly selected and approximately 10 mL of blood was collected by exsanguination from the jugular vein, and serum was separated by centrifuging at 4000× *g* for 10 min and then stored at −80 °C for subsequent analysis. After blood collection, the broilers were killed by cervical dislocation. The whole thigh (*biceps femoris*) and breast (*pectoralis major*) muscle tissues from the same carcass were removed. One side of the thigh and breast muscle stored at 4 °C was used for meat quality trait analysis. Another side of the thigh and breast muscle stored at −80 °C was used to for inosine monophosphate (IMP) and fatty acid composition analysis. Samples of liver were obtained and stored at −80 °C until analysis. Cecal contents were also collected for 16S rRNA gene sequencing and metabolite analysis.

### 2.3. Sample Processing

Muscle samples of approximately 80 mg were homogenized with 1 mL of extraction solution (acetonitrile: water = 1:1, *v*/*v*) for 2 min. The samples were then sonicated in an ice-water bath for 3 min, and centrifuged at 12,000× *g* for 15 min at 4 °C. Each supernatant was transferred to an autosampler vial, and 5.0 μL was injected to the ultrahigh-performance liquid chromatography-tandem mass spectrometry (UPLC–MS) system for IMP analysis.

Muscle samples of approximately 50 mg were homogenized with 1mL extraction solution (isopropanol: n-hexane = 2:3, *v*/*v*) for 4 min. The samples were then sonicated in an ice-water bath for 3 min, and centrifuged at 12,000× *g* for 15 min at 4 °C. After drying the supernatant in a vacuum concentrator (Memmert, Germany), the residue was dissolved in methanol (1.0 mL) and then trimethylalkyldizomethane (5.0 mL, 2.0 M) was added to the mixture for methyl esterification step. After standing at room temperature for 15 min, the excess silylating reagent was evaporated by nitrogen blowdown, and the sample mixture was added with 2 mL of n-hexane. Finally, 2.0 μL of the supernatant was injected into the gas chromatography-mass spectrometry (GC–MS) system for fatty acid composition analysis.

Cecal content samples of approximately 100 mg were prepared via protein precipitation with 500 µL of methanol. Then the samples were vigorously vortexed for 5 min, and centrifuged at 20,000× *g* for 10 min at 4 °C. After the supernatant was collected, the residue again was extracted according to the above extraction procedure and combined with the previous supernatant. At last, 200 µL of supernatant was transferred into the sampling vial for 16S rRNA analysis.

Cecal content samples and AGE of approximately 50 mg were added into an Eppendorf tube with 1000 μL extractant solution (methanol: acetonitrile: water = 2:2:1, *v*/*v*) and 1 mg/mL internal standard, respectively. Then, the samples were homogenized at 35 Hz for 4 min and sonicated for 5 min on ice. The homogenization and sonication cycle were repeated for 3 times. After that, the samples were incubated for 1 h at −40 °C and centrifuged at 12,000× *g* for 15 min at 4 °C. The resulting supernatant was transferred to a spotless glass vial for metabolomics analysis as previously described [23]. The quality control (QC) sample was prepared by mixing an equal aliquot of the supernatants from all of the samples.

Liver samples of approximately 100 mg were homogenized in 5 mL of PBS (pH 7.4). Following centrifugation at 2500× *g* for 20 min at 4 °C, the resulting supernatant was used in the subsequent enzyme activity analysis.

### 2.4. Serum Lipids Parameter Analysis

The concentrations of total cholesterol (TCHO), total triglyceride (TG), low density lipoprotein cholesterol (LDL-C) and high-density lipoprotein cholesterol (HDL-C) in serum samples were analyzed by an automatic biochemical analyzer (RA-1000, Bayer Corp., Tarrytown, NY, USA) following the instructions of the commercial kits (Zhongsheng Biochemical Co., Ltd., Beijing, China).

### 2.5. Meat Quality Analysis

#### 2.5.1. Meat Quality Traits

The preparation of meat samples and the measurement method of meat quality were as previously described [24]. The pH value of thigh and breast muscle was measeured approximately 45 min and 24 h postmortem, using a penetrating electrode (Mettler Toledo, Changzhou, China) attached to a portable pH-meter (FG2, Shanghai, China) into the *biceps femoris* and *pectoralis major* muscle. The pH probe was calibrated with pH 4 and pH 7 standard buffer solutions before operation.

The color measurements of meat (L* = relative lightness; a* = relative redness; and b* = relative yellowness) were measured with a hand-held color reader (CR 10, Konica Minolta INC., Osaka, Japan). A reading was made from the inner surface of sample, representing the whole surface of the muscle. A white tile (L* 92.30, a* 0.32, and b* 0.33) was used as standard.

Approximately 20 g of samples from the thigh and breast muscles were used to determine the drip loss. The muscle samples were weighed, placed in a plastic bag filled with air and fastened to avoid evaporation, suspended in a 4 °C cooler for 24 h and then reweighed. Drip loss was calculated based on the weight loss and was expressed as a percentage.

After the pH value and drip loss measurements, the thigh and breast muscle samples were packed into plastics bags and brought to room temperature before cooking. Then, the muscle samples were cooked to an internal temperature of 70 °C in a thermostatic water-bath. Cooked muscle was cooled to room temperature. They were then cut into 3 × 1 × 1 cm^3^ pieces parallel to the orientation of the muscle fiber. Warner–Bratzler shear force was assessed using a Texture Measurement System (Food Technology Corporation, Stirling, VA, USA).

#### 2.5.2. Inosine Monphosphate Analysis

The LC–MS analysis was performed using an UltiMate™ 3000 ultra-high performance liquid chromatography (UPLC) system (Thermo Scientific, Waltham, MA, USA) coupled to a QExactive^TM^-Plus Orbitrap (Thermo Scientific, Waltham, MA, USA) mass spectrometer. Chromatographic separation was achieved using an EC-C18 column (100 mm × 2.1 mm, 2.7 μm; Agilent). The mobile phase consisted of solvent A (acetonitrile) (AR, Thermo Scientific, Waltham, MA, USA), and solvent B (water with 0.3% formic acid) (AR, Thermo Scientific, Waltham, MA, USA). The gradient elution was used at a flow rate of 0.2 mL/min as follows: between 0 to 6 min, the linear gradient is from 2.0% A to 6.6% A; and between 6.1 to 11 min, the linear remains at 2% A. The column oven was maintained at 40 °C and the target mass range was 70–1000 *m*/*z,* applying an MS reflector in a positive ion mode. The MS parameters were as follows: a probe heater temperature of 300 °C; a capillary temperature of 320 °C; 40 psi sheath gas; 15 psi aux gas; and a spray voltage of 3.2 kV.

#### 2.5.3. GC–MS Analysis of Fatty Acids

The GC–MS analysis was performed following the methodology previously described, with minor modifications [25]. Briefly, an aliquot of 2 μL of the derivatized sample was injected in splitless mode on an Agilent 7890 B gas chromatograph coupled with an Agilent 5977 A mass spectrometer. Separation was performed on an HP-5MS capillary column (30 m × 0.25 mm × 0.25 μm film thickness, (Agilent, Palo Alto, USA) with He as the carrier gas at a constant flow rate of 0.8 mL/min. The solvent delay time was set to 2.5 min. The temperatures of injection, transfer interface and ion source were 230 °C, 240 °C and 230 °C, respectively. The GC temperature programming was set to 2 min of isothermal heating at 60 °C, followed by 13 °C/min with the oven temperature increased to 150 °C and 2 °C/min to 230 °C, and a final 6 min maintenance at 230 °C. Ionization mode was electron ionization (EI) mode with the electron energy of 70 eV. A full scan mode with a scanning range of *m*/*z* 50 to 450 was used. Fatty acid methyl esters were identified by comparing retention times to known standards and by matching them up with what is provided in the mass spectra profile held in the NIST 2014 Spectral Library. Then the content of fatty acid methyl esters was quantified in a targeted approach using Agilent’s Mass Hunter Quantitative Analysis software.

### 2.6. 16S rRNA-Based Sequencing Analysis

Total genome DNA from each sample was extracted using the CTAB/SDS method [26] and diluted to 1 ng/μL. The bacterial 16S rRNA was amplified using the V3-V4 region primer 515F (5′-GTGCCAGCMGCCGCGGTAA-3′) and 806R (5′-GGACTACHVGGGTWTCTAAT-3′). PCR-amplification products were detected by 2% agarose gel electrophoresis and were purified with a Gel Extraction Kit (Qiagen, Germany). Following this, both library concentration and an exact product size were measured using a TruSeq DNA PCR-Free Sample Preparation Kit through a quantitative PCR and Qubit. The PCR-free library was constructed based on the illumina NovaSeq platform.

Raw fastq files were quality-filtered using Trimmomatic and merged using FLASH (Version 1.2.7, http://ccbjhu.edu/saftware/FLASH/, accessed on 15 May 2022). The raw data from 16S rRNA amplicon sequencing is filtered through QIIME (Version 1.9.1, http://qiime.org/scriDts/split_libraries_fkstq.html, accessed on 15 May 2022) quality control process to obtain high-quality tags. Operational taxonomic units (OTUs) were clustered with a 97% threshold using UPARSE (Version 7.1, http://drive5.com/uparse/, accessed on 15 May 2022). Multiple sequence alignment was conducted using the MUSCLE software (Version 3.8.31, https://drive5.com/muscle, accessed on 15 May 2022). Alpha diversity (Chao 1, ACE, Shannon, Simpson, Observed-species, Good-coverage) was calculated with QIIME (Version 1.9.1), and beta diversity (weighed and unweighted UniFrac) was estimated with QIIME (Version 1.9.1). Both biodiversities were displayed with R software (Version 2.15.3, http://www.r-project.org/, accessed on 15 May 2022).

### 2.7. Metabolomics Analysis of Cecal Contents and Metabolic Profile of Dihydromyricetin and Myricetin from AGE

LC–MS/MS analyses were performed using an UPLC system (Thermo Scientific, Waltham, MA, USA) with a UPLC BEH Amide column (2.1 mm × 100 mm, 1.7 μm, Waters, Milford, MA, USA), coupled with a Q Exactive^TM^ HFX mass spectrometer (Orbitrap MS, Thermo Scientific, Waltham, MA, USA). Mobile phase A consisted of 25 mmol/L ammonium acetate (AR, Thermo Scientific, Waltham, MA, USA) and 25 mmol/L ammonia hydroxide (AR, Thermo Scientific, Waltham, MA, USA) in water (pH = 9.75). Mobile phase B consisted of acetonitrile (AR, Thermo Scientific, Waltham, MA, USA). The auto-sampler temperature was 4 °C, and the injection volume was 3 μL.

The QE HFX mass spectrometer was used for its ability to acquire MS/MS spectra on information-dependent acquisition (IDA) mode in the control of the acquisition software (Xcalibur, Thermo). In this mode, the acquisition software continuously evaluates the full scan MS spectrum. The ESI source conditions were set as following: sheath gas flow rate as 30 Arb, aux gas flow rate as 25 Arb, capillary temperature 350 °C, full MS resolution as 60,000, MS/MS resolution as 7500, collision energy as 10/30/60 in NCE mode and spray voltage as 3.6 kV (positive) or −3.2 kV (negative) [23].

The raw data from fecal samples were converted to the mzXML format using ProteoWizard and processed with an in-house program, which was developed using R and based on XCMS, for peak detection, extraction, alignment and integration. Then, an in-house MS^2^ database (Biotree DB, Shanghai, China) was applied in metabolite annotation. The cutoff for annotation was set at 0.3. In this study, 1055 peaks were detected, and 944 metabolites remained after de-noising relative to the standard deviation and the missing values were filled by half of the minimum value. To visualize the metabolic difference among control group and AGE groups, the principal component analysis (PCA) and partial least squares discrimination analysis (PLS-DA) models were carried out. Furthermore, the value of fold change (FC) was calculated as the average normalized peak intensity ratio between the two groups. Differences between data sets with FC > 1.10 or FC < 0.90and *p* < 0.05 (Student’s *t*-test) were considered statistically significant.

In addition, metabolic profiling of dihydromyricetin and myricetin in intestinal tissue were analyzed by UPLC–MS/MS with Compound Discoerer 3.1 software (Thermo Scientific Inc., Waltham, MA, USA).

### 2.8. ELISA Analysis

The activity of 3-hydroxy-3-methylglutaryl-CoA reductase (HMGR) and Cholesterol 7α Hydroxylase (CYP7A1) in the liver were measured using ELISA kits (Shanghai Huyu biotechnology Co., Ltd., Shanghai, China) according to the manufacturer’s instructions.

### 2.9. Statistical Analysis

Statistical analysis of serum lipids parameters, meat quality and the profile of fatty acids, among the groups, were evaluated using the one-way analysis of variance (ANOVA), performed by SPSS 23.0 (IBM-SPSS Inc., Chicago, IL, USA). The results in the tables are presented as mean and pooled SEM, and orthogonal polynomial contrasts were used to test the linear and quadratic effects of AGE. Other figure results were shown with means ± SD. Significant differences were evaluated by Tukey’s multiple comparison test at *p* < 0.05.

## 3. Results

### 3.1. Serum Lipids Parameters

The data of the serum parameters are shown in Table 2. The levels of LDL-C and TG are linearly reduced (*p* < 0.05) with dietary AGE supplementation. Meanwhile, dietary AGE supplementation at 0.2% and 0.4% reduces the TCHO concentration significantly (*p* < 0.05) compared with the T1 group.

### 3.2. Meat Quality Traits

The effects of AGE dietary supplementation on meat traits are shown in Table 3. The T3 and T4 groups had a higher (*p* < 0.05) a* value in thigh and breast muscles than the T1 group. Additionally, the dietary supplementation of AGE linearly decreased the shear force and drip loss in both the thigh and breast muscles (*p* < 0.05). Meanwhile, compared with the T1 group, dietary AGE supplementation at 0.2% and 0.4% increased the levels of IMP significantly (*p* < 0.05) both in thigh and breast muscles.

### 3.3. Fatty Acids

The effects of AGE addition on fatty acid composition in thigh and breast muscles of the broilers are presented in Table 4 and Table 5. The thigh and breast muscle in the T3 and T4 groups have lower (*p* < 0.05) saturated fatty acid (SFA) (especially C14:0 and C16:0) levels than the T1 group (also T2 in breast), while the polyunsaturated fatty acid (PUFA) (mainly C22:6n3) content in the T3 groups is increased (*p* < 0.05) compared with the T1 group. The T4 group also resulted in an increase (*p* < 0.05) of C16:1n7 in breast muscle, whereas a significant increase (*p* < 0.05) of C20:1n9 in breast muscle is observed in the T3 group. Furthermore, a linearly increase (*p* < 0.05) of the total unsaturated fatty acid (USFA), monounsaturated fatty acids (MUFA), PUFA and the ratio of USFA to SFA (USFA: SFA) both in thigh and breast muscles in AGE-treated groups is observed.

### 3.4. Microbial Composition

As shown in Appendix A, the dilution curves and rank abundance curves indicate that the number of samples was sufficient to satisfy the sequencing depth. Alpha diversity analysis included the observed species, as well as Shannon, Chao 1, ACE and Simpson indexes are shown in Appendix A.

The relative abundance of taxa observed at the phyla and genus levels between the T1 and T3 group are illustrated in Figure 1. When assessing the microbiota composition at the phylum level, we found that *Bacteroidota*, *Firmicutes* and *Actinobacteriota* were the dominant microbes in both groups of individuals. Higher abundance of *Bacteroidota* (*p* < 0.05) and lower abundance of *Firmicutes* (*p* < 0.05) were observed in the T3 group. Moreover, at the genus level, *Bacteroides*, and *Rikenellaceae_RC9_gut_group* were the main genera in both groups. Furthermore, the abundance of *Faecalibacterium* was decreased (*p* < 0.05) in the T3 group compared with the T1 group.

### 3.5. Metabolomics Analysis

To investigate the different metabolic pathway in the cecal contents between the T1 group and T3 groups, all observations were integrated and analyzed using PCA. As observed in Appendix A, the PCA analysis showed that the TI and T3 groups were separated in the PC1 × PC2 score plots with 27.5% of PC1 and 20.5% of PC2. Supervised PLS-DA was applied to further explore the differences between the two groups. As shown in Appendix A, the score plots of PLS-DA suggested that the two groups could be clearly separated, which reflected the remarkably distinct metabolic status of the cecal samples between the T1 and T3 group.

As observed in Figure 2 and Appendix A, p-cresol sulfate and cholesterol sulfate were identified as differential metabolites between the T1 and T3 groups. It suggested that AGE supplementation at 0.4% significantly downregulated the levels of p-cresol sulfate and cholesterol sulfate (*p* < 0.05) compared with the T1 group.

### 3.6. Correlation Analysis between Microbiota and Metabolites in the Cecal Content

We further analyzed the correlation between metabolites and microbiota in the cecal content between the T1 and T3 groups (Figure 3). The result showed that the level of p-cresol sulfate positively correlated with *Faecalibacterium* (*p* < 0.05).

### 3.7. Metabolic Profile of Dihydromyricetin and Myricetinin from the AGE

The metabolites of dihydromyricetin and myricetin in cecal content were investigated. The retention time (RT), metabolic pathways, neutral formulas and fragment ions of these metabolites are summarized in Appendix A. It was found that metabolite M1 (RT = 1.004 min) is a metabolite detected simultaneously in the metabolites of both dihydromyricetin and myricetin. The metabolic pathways of metabolite M1 were described as follows, respectively.

The proposed fragmentation pathways of dihydromyricetin are shown in Figure 4A. The metabolite M1 was identified with an accurate [M−H]^−^ ion at *m*/*z* 303.0406, which was 16 Da less than that of the parent compound, implying the dehydration and reduction in dihydromyricetin. The MS^2^ fragmentation generated the fragment ions at *m*/*z* 125.0246, 151.0034, 181.0159 and 285.0560. The ion at *m/z* 285.0560 was produced by the loss of a water molecule, and the product ion at *m*/*z* 151.0034 was generated because of the RDA (Retro Diels-Alder reaction) fragmentation. The bond cleavage between C4−C10 and C2−O1 produced the product ions at *m*/*z* 125.0243 and 177.0192.

The proposed fragmentation pathways of myricetin are shown in Figure 4B. The metabolite M1 was identified with an accurate [M−H]^−^ ion at *m*/*z* 303.0406, which was 14 Da less than that of the parent compound, implying the reduction and oxidation of myricetin. The MS^2^ fragmentation generated the fragment ions at *m*/*z* 125.0238, 151.0034, 183.9984 and 285.0612. The product ion at *m*/*z* 285.0612 was produced by loss of a water molecular, and the product ion at *m*/*z* 151.0034 could be generated because of the RDA fragmentation. The bond cleavage between C4–C10 and C2–O1 produced ions at *m*/*z* 125.0238 and m/z 183.9984.

According to the present findings, the metabolite M1 is tentatively identified as dihydroquercetin.

### 3.8. ELISA Analysis

The liver is the main site of fat metabolism in poultry. To further explore the effects of treatment with AGE on lipid metabolism in broilers, we mainly analyzed the levels of the key hepatic fat metabolism synthase HMGR and CYP7A1 between the T1 group and the T3 group (Figure 5). The hepatic HMGR enzyme activity in the T3 group was lower (*p* < 0.05) than the T1 group.

## 4. Discussion

In the present study, our results show that the dietary supplementation of AGE quadratically decreased serum TCHO, linearly reduced TG and LDL-C and linearly increased serum HDL-C concentrations in broilers, which suggests that flavonoid in the AGE might be involved in regulation of lipid metabolism [27]. Particularly, dietary supplementation of AGE at 0.4% could provide beneficial effects in contributing to the improvement of TCHO, TG, LDL-C and HDL-C contents in the serum of broilers. Then, the activity of hepatic HMGR, which is the key enzyme of cholesterol biosynthesis [28], was further assessed between the T1 and T3 groups. It was found that the dietary supplementation of AGE at 0.4% markedly inhibited HMGR enzyme activity and thereby reduced the synthesis of cholesterol in the liver. This may eventually lead to reductions of serum TCHO and cholesterol sulfate concentrations in the cecal content, which is synthesized from cholesterol by steroid sulfatase or alcohol sulfotransferase. As it is known, the activation state of hepatic HMGR enzyme is influenced by the increase in reactive oxygen species (ROS) [29]. Therefore, we speculate that HMGR inhibition is related to the antioxidant activity of the high content of flavonoids in the AGE [30,31].

Generally, meat quality should be reflected by several indexes, such as meat color, drip loss, shear force and intramuscular fat content. Poultry meat color is an important food quality index and a higher a* values of meat is always favored by customers. In this study, the T3 and T4 groups had higher a* value in both the thigh and breast muscles than the T1 group. Additionally, the dietary supplementation of AGE linearly decreased the shear force in both the thigh and breast muscles than the T1 group. It is well known that the molecular structure of flavonoids contains phenolic hydroxyl groups which can provide hydrogen atoms to neutralize free radicals, making it a strong antioxidant [32]. It is proposed that, flavonoid-rich AGE as an antioxidant supplement might not only effectively prevent deoxymyoglobin (red color) from being oxidized to metmyoglobin (brown color) thus increasing the a* value in meat [33], but may also affect muscle protein degradation, which causes some decrease in drip loss and the shear force of meat [34,35]. IMP is an endogenous compound in muscle. It is well known to affect meat flavor and IMP contributes to the umami taste, alone or conjugated with monosodium glutamate for synergistic effects. Therefore, increasing IMP concentration in meat may improve the sensory quality of meat [36,37]. It was found that AGE supplementation could significantly enhance the content of IMP in both thigh and breast muscle of broilers, which contributes to the taste and flavor of meat. On the other hand, a significant increase in PUFA concentration in the T3 groups compared with the T1 group was also observed. The primary reason could be that flavonoids in AGE possess antioxidant properties which might help protect PUFA from oxidation [38,39]. Although the content of total PUFA was increased in the T2 group compared with the T1 group, no significant difference was observed between the T1 and T2 groups. This may be related to the fact that the T2 group is characterized by the lowest flavonoid content among the three AGE groups. In addition, diet supplementation with AGE enhanced the ratio of USFA to SFA and decreased the concentration of C14:0 and C16:0 compared with the T1 group, which are beneficial for both improving the quality of nutrition and preventing the development of cholesterolemia [40,41,42]. Furthermore, it revealed that AGE supplementation could effectively regulate blood lipid through increasing the content of the ω-3 PUFAs, including C20:5n-3 and C22:6n-3 compared with the T1 group [42].

Additionally, *Bacteroidota* and *Firmicutes* can regulate intestinal immune system by affecting the release of proinflammatory cytokines [43]. As shown in our result, microbiome composition in both the T1 and T3 groups mainly consist of *Bacteroidota*, *Firmicutes*, *Actinobacteriota* and *Proteobacteria*. This result is consistent with previous studies [44]. Compared with the T1 group, the relative abundance of *Bacteroidota* in the T3 group markedly increased, while the relative abundance of *Firmicutes* significantly decreased. Previous research showed that the abundance of *Bacteroidetes* correlated positively with the efficiency of lipid hydrolysis, that is, the increased abundance of *Bacteroides* contributed to a higher colipase activity [45]. Furthermore, the abundance of *Firmicutes* is positively correlates with the generation of metabolic endotoxins and thus a lower *Firmicutes* content has been considered beneficial to prevent inflammation in the gut [46]. Moreover, the results of our analysis indicate that a lower level of p-cresol sulfate was found to be positively linked with the decreased abundance of *Faecalibacterium*. Meanwhile, our data reveals that the level of p-cresol sulfate in the T3 group was decreased compared with the T1 group, which suggests that AGE supplements might inhibit oxidative damage in the intestine [31].

In addition, the present study further investigates the intestinal metabolism of dihydromyricetin and myricetin from AGE. Notably, it shows that dihydroquercetin is a common metabolite generated by both dihydromyricetin and myricetin. It has been reported that dihydroquercetin exhibit antioxidant effect probably due to presence of the flavonoid structure [47].

## 5. Conclusions

Our present study has demonstrated that dietary supplementation with 0.2% to 0.6% of AGE can increase the IMP and polyunsaturated fatty acid content in the muscle of Wenchang broilers. Dietary supplementation with 0.4% of AGE can reduce the abundance of *Faecalibacterium* and down-regulate the concentration of p-cresol sulfate in the cecal content of Wenchang broilers. Furthermore, dietary AGE supplementation with 0.4% can regulate the cholesterol metabolism by decreasing the concentration of p-cresol sulfate in the cecal content and reducing the hepatic HMGR enzyme activity of Wenchang broilers. Metabolism model of myricetin and dihydromyricetin, the two main flavonoids in AGE, indicate that both of them can be metabolized into dihydroquercetin in the intestine of chickens.

## Figures and Tables

**Figure 1 animals-12-01661-f001:**
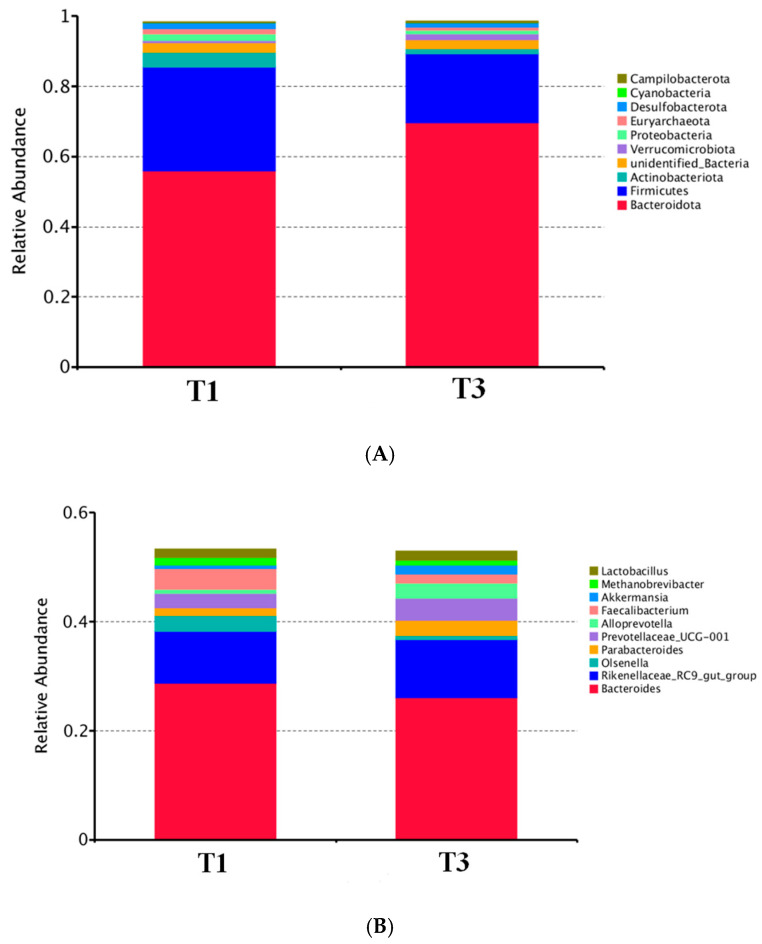
Microbial composition of cecum contents of broilers at phylum (**A**) and genus (**B**) levels. T1: the broilers were fed a basal diet; T3: the broilers were fed a basal diet supplemented with AGE at 0.4%.

**Figure 2 animals-12-01661-f002:**
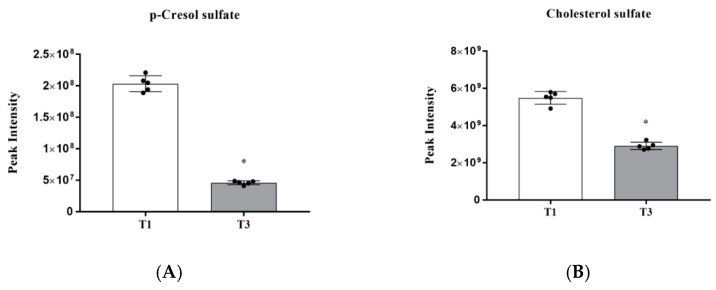
Differential metabolites from T1 group vs. T3 group. (**A**) p-Cresol sulfate and (**B**) Cholesterol sulfate. T1: the broilers were fed a basal diet; T3: the broilers were fed a basal diet supplemented with AGE at 0.4%. * *p* < 0.05.

**Figure 3 animals-12-01661-f003:**
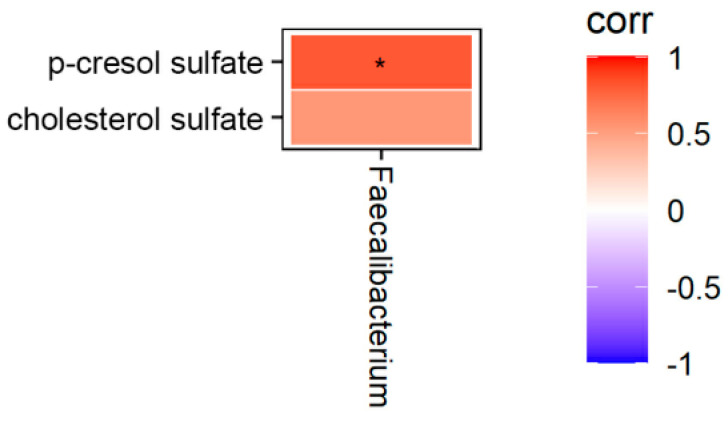
The correlation analysis between metabolites and microbiota in the cecal content. Each row represents a metabolite with color coding that indicates the correlation between fecal metabolites and the gut microbiota. The correlation of each metabolite was normalized in the range of −1 to 1. Blue (−1) and red (1) represent the lowest and highest levels, respectively. * *p* < 0.05.

**Figure 4 animals-12-01661-f004:**
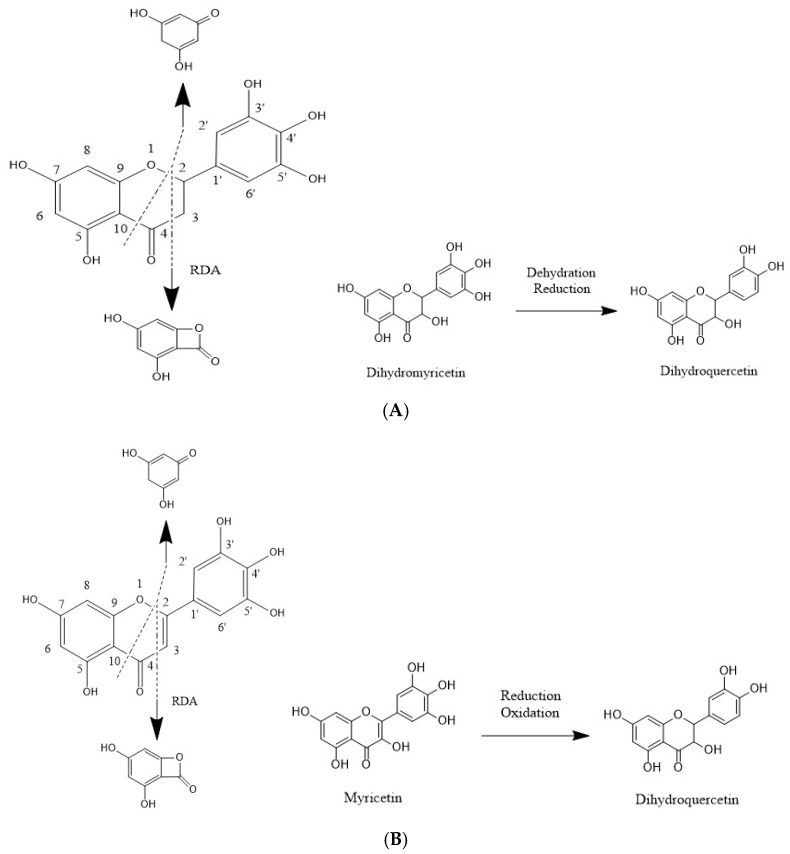
The proposed fragmentation pathways of the metabolite M1 from dihydromyricetin (**A**) and myricetin (**B**), respectively.

**Figure 5 animals-12-01661-f005:**
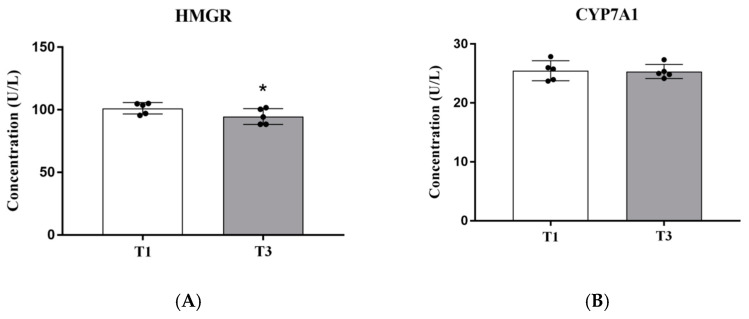
Effects of AGE supplementation on the enzyme activities of HMGR (**A**) and CYP7A1 (**B**) in the liver of broilers. * *p* < 0.05. T1: the broilers were fed a basal diet; T3: the broilers were fed a basal diet supplemented with AGE at 0.4%. HMGR,3-hydroxy-3-methylglutaryl-CoA reductase; CYP7A1, cholesterol 7α-hydroxylase.

**Table 1 animals-12-01661-t001:** Ingredient and nutrient levels of basal diet (%, as-fed).

Items	Contents
Ingredients	
Corn	63.55
Fermented soybean meal, CP ≥ 46%	11.60
Middling, CP ≥ 15%	5.00
Corn gluten meal, CP ≥ 60%	5.00
Rapeseed meal, CP ≥ 38%	3.00
Rice bran meal, CP ≥ 15%	1.50
Soybean oil	7.00
Stone powder	0.50
Calcium hydrogen phosphate	0.40
Premix ^1^	2.45
Total	100
Nutrient levels ^2^	
Metabolic energy (MJ/kg)	13.96
Crude protein	15.32
Lysine	0.58
Methionine	0.27
Cysteine	0.27
Calcium	0.62
Available phosphorus	0.44

^1^ Each kilogram of premix contains: VA 1,000,000 IU, VD_3_ 416,667 IU, VE 6667 mg, VK_3_ 267 mg, VB_1_ 267 mg, VB 2717 mg, VB_6_ 450 mg, nicotinamide 5000 mg, calciumpantothenate 1417 mg, Fe 1667 mg, Cu 1333 mg, Mn 10,000 mg, Zn 9167 mg, I 104 mg, Se 25 mg. ^2^ The nutrient levels are calculated values.

**Table 2 animals-12-01661-t002:** Effects of *Ampelopsis grossedentata* extract on the serum lipids parameter of broilers.

Items	Groups				SEM	*p*	Contrast	
	T1	T2	T3	T4			*L*	*Q*
TCHO (mmol/L)	4.01 ^a^	3.53 ^b^	3.45 ^b^	3.74 ^ab^	0.11	0.012	0.093	0.003
LDL-C (mmol/L)	1.37 ^a^	1.21 ^ab^	1.10 ^b^	1.12 ^b^	0.06	0.014	0.003	0.130
HDL-C (mmol/L)	2.21	2.40	2.41	2.47	0.07	0.077	0.019	0.353
TG (mmol/L)	0.62 ^a^	0.60 ^ab^	0.53 ^b^	0.54 ^b^	0.02	0.018	0.004	0.594

Values in the same row with no letter or the same letter superscripts mean no significant difference (*p* > 0.05), while with different letter superscripts mean significant difference (*p* < 0.05).T1: the broilers were fed a basal diet; T2: the broilers were fed a basal diet supplemented with AGE at 0.2%; T3: the broilers were fed a basal diet supplemented with AGE at 0.4%; T4: the broilers were fed a basal diet supplemented with AGE at 0.6%.TCHO: total cholesterol; LDL-C: low density lipoprotein cholesterol; HDL-C: high density lipoprotein cholesterol; TG: triglycerides.

**Table 3 animals-12-01661-t003:** Effects of *Ampelopsis grossedentata* extract on meat quality of broilers.

Item	Groups				SEM	*p*	Contrast	
	T1	T2	T3	T4			*L*	*Q*
Thigh muscle								
*L* *	38.4	38.54	37.26	37.4	0.98	0.715	0.343	1.000
a *	15.02 ^b^	15.90 ^ab^	18.20 ^a^	18.54 ^a^	0.80	0.016	0.002	0.739
b *	11.58	10.5	10.48	9.86	0.57	0.230	0.057	0.690
pH_45min_	6.28	6.54	6.40	6.28	0.07	0.055	0.698	0.015
pH_24h_	5.91	5.98	6.05	5.95	0.04	0.179	0.360	0.067
Shear force (N)	24.83 ^a^	21.75 ^b^	21.55 ^b^	21.80 ^b^	0.62	0.005	0.004	0.016
Drip loss (%)	5.46 ^a^	5.21 ^ab^	4.93 ^b^	5.00 ^b^	0.10	0.010	0.003	0.143
IMP (mg/g)	3.08 ^b^	3.79 ^a^	3.73 ^a^	3.38 ^ab^	0.14	0.008	0.201	0.002
Breast muscle								
*L**	48.64	48.42	47.28	46.70	0.54	0.065	0.011	0.743
a*	4.18 ^b^	4.60 ^ab^	5.00 ^a^	4.82 ^a^	0.13	0.003	0.001	0.037
b*	15.88	16.40	13.88	14.16	0.70	0.051	0.025	0.865
pH_45min_	6.32	6.15	6.01	6.03	0.12	0.271	0.076	0.429
pH_24h_	5.55	5.54	5.60	5.50	0.03	0.230	0.586	0.169
Shear force (N)	16.95 ^a^	14.70 ^b^	14.35 ^b^	13.87 ^b^	0.60	0.011	0.003	0.159
Drip loss (%)	5.11 ^a^	4.48 ^ab^	4.25 ^b^	4.49 ^ab^	0.17	0.016	0.015	0.022
IMP (mg/g)	2.22 ^b^	3.14 ^a^	3.12 ^a^	2.79 ^ab^	0.19	0.011	0.064	0.005

Values in the same row with no letter or the same letter superscripts mean no significant difference (*p* > 0.05), while with different letter superscripts mean significant difference (*p* < 0.05). T1: the broilers were fed a basal diet; T2: the broilers were fed a basal diet supplemented with AGE at 0.2%; T3: the broilers were fed a basal diet supplemented with AGE at 0.4%; T4: the broilers were fed a basal diet supplemented with AGE at 0.6%.IMP, inosine monophosphate.

**Table 4 animals-12-01661-t004:** Fatty acids profile in the thigh muscle of broilers (%).

Items	Groups				SEM	*p*	Contrast	
	T1	T2	T3	T4			L	Q
C6:0	4.43	4.61	4.45	4.59	0.07	0.202	0.344	0.756
C8:0	2.47	2.57	2.71	2.39	0.10	0.174	0.863	0.053
C10:0	5.22	5.42	5.38	5.52	0.09	0.144	0.040	0.737
C11:0	4.40	4.30	4.58	4.59	0.09	0.106	0.052	0.576
C14:0	14.6 ^a^	14.16 ^ab^	13.75 ^b^	13.61 ^b^	0.43	0.002	0.001	0.405
C16:0	0.83 ^a^	0.78 ^ab^	0.77 ^b^	0.74 ^b^	0.02	0.006	0.001	0.473
C17:0	6.50	6.42	6.68	6.46	0.09	0.249	0.737	0.472
C18:0	7.57	7.43	7.16	7.24	0.16	0.302	0.098	0.510
C20:0	0.16	0.17	0.12	0.14	0.01	0.073	0.076	0.911
C21:0	0.37	0.36	0.33	0.34	0.01	0.136	0.033	0.567
C22:0	1.17±	1.14	0.99	1.13	0.05	0.065	0.221	0.079
C14:1n5	1.47	1.53	1.67	1.66	0.08	0.253	0.065	0.660
C15:1n5	0.28	0.23	0.26	0.21	0.02	0.043	0.038	0.842
C16:1n7	9.18	9.19	9.63	9.50	0.21	0.362	0.158	0.745
C17:1n7	0.25	0.21	0.27	0.22	0.02	0.102	0.785	0.817
C18:1n9	21.26	21.13	21.71	21.79	0.23	0.153	0.051	0.662
C20:1n9	0.10	0.13	0.12	0.15	0.02	0.285	0.131	0.868
C22:1n9	0.26	0.25	0.27	0.29	0.01	0.072	0.022	0.223
C18:2n	5.75	5.36	5.94	5.82	0.17	0.142	0.322	0.441
C18:3n6	7.84	7.95	8.09	7.81	0.11	0.329	0.905	0.109
C20:2	1.04	1.00	0.95	0.97	0.04	0.444	0.173	0.462
C20:3n6	1.02	1.06	1.08	1.05	0.02	0.357	0.324	0.142
C20:4n6	4.24	4.22	4.28	4.32	0.04	0.353	0.122	0.473
C20:5n3	2.40 ^b^	2.51 ^b^	2.80 ^a^	2.55 ^b^	0.06	0.001	0.008	0.006
C22:2n6	0.99	1.08	1.00	1.05	0.03	0.135	0.408	0.495
C22:6n3	1.03 ^b^	1.23 ^ab^	1.32 ^a^	1.37 ^a^	0.07	0.015	0.002	0.296
SFA	47.72 ^a^	47.35 ^a^	46.94 ^b^	46.74 ^b^	0.34	<0.001	<0.001	0.394
USFA	57.10 ^b^	57.09 ^b^	59.39 ^a^	58.78 ^a^	0.34	<0.001	<0.001	0.394
MUFA	32.79	32.68	33.93	33.83	0.32	0.019	0.007	0.987
PUFA	24.31 ^b^	24.41 ^b^	25.46 ^a^	24.96 ^ab^	0.27	0.030	0.025	0.282
USFA/SFA	1.20 ^b^	1.21 ^b^	1.27 ^a^	1.26 ^a^	0.02	<0.001	<0.001	0.375

Values in the same row with no letter or the same letter superscripts mean no significant difference (*p* > 0.05), while with different letter superscripts mean significant difference (*p* < 0.05).T1: the broilers were fed a basal diet; T2: the broilers were fed a basal diet supplemented with AGE at 0.2%; T3: the broilers were fed a basal diet supplemented with AGE at 0.4%; T4: the broilers were fed a basal diet supplemented with AGE at 0.6%.SFA, saturated fatty acid; USFA, unsaturated fatty acids; MUFA, monounsaturated fatty acids; PUFA, polyunsaturated fatty acids; USFA: SFA, the ratio of unsaturated fatty acids to saturated fatty acid.

**Table 5 animals-12-01661-t005:** Fatty acids profile in the breast muscle of broilers (%).

Items	Groups				SEM	*p*	Contrast	
	T1	T2	T3	T4			L	Q
C6:0	4.25	3.91	3.99	3.95	0.11	0.136	0.095	0.180
C8:0	2.15	2.30	1.99	2.31	0.10	0.110	0.738	0.412
C10:0	5.16	5.16	4.94	5.08	0.15	0.706	0.506	0.637
C11:0	4.98	4.99	4.94	4.96	0.04	0.877	0.576	0.916
C14:0	14.86 ^a^	13.73 ^b^	13.39 ^b^	13.18 ^b^	0.18	<0.001	<0.001	0.023
C16:0	0.82 ^a^	0.78 ^ab^	0.75 ^b^	0.74 ^b^	0.01	0.013	0.002	0.272
C17:0	2.27	2.26	2.80	2.46	0.20	0.239	0.236	0.429
C18:0	4.64	4.54	4.31	4.47	0.11	0.212	0.137	0.256
C20:0	0.15	0.16	0.17	0.13	0.01	0.109	0.258	0.050
C21:0	0.18	0.17	0.11	0.12	0.02	0.046	0.015	0.662
C22:0	1.44	1.37	1.38	1.40	0.02	0.060	0.223	0.023
C14:1n5	1.37	1.36	1.43	1.39	0.03	0.481	0.347	0.628
C15:1n5	0.27	0.25	0.20	0.21	0.02	0.118	0.025	0.569
C16:1n7	9.90 ^b^	10.18 ^ab^	10.19 ^ab^	10.67 ^a^	0.18	0.047	0.009	0.576
C17:1n7	0.20	0.21	0.15	0.16	0.02	0.195	0.098	0.928
C18:1n	17.85	17.97	18.33	17.83	0.17	0.193	0.704	0.091
C20:1n9	0.13 ^b^	0.15 ^ab^	0.19 ^a^	0.16 ^ab^	0.01	0.018	0.032	0.061
C22:1n9	0.32	0.35	0.36	0.31	0.01	0.060	0.737	0.011
C18:2n	6.45	6.56	6.75	6.63	0.16	0.601	0.319	0.482
C18:3n6	9.84	9.79	9.96	9.74	0.13	0.689	0.845	0.535
C20:2	1.32	1.39	1.36	1.38	0.03	0.465	0.291	0.510
C20:3n6	1.43	1.36	1.45	1.35	0.04	0.216	0.383	0.677
C20:4n6	4.54	4.23	4.42	4.19	0.14	0.268	0.180	0.768
C20:5n3	2.13 ^b^	2.52 ^ab^	2.67 ^a^	2.47 ^ab^	0.38	0.108	0.134	0.777
C22:2n6	0.90	0.94	0.92	0.89	0.03	0.528	0.764	0.201
C22:6n3	1.01 ^b^	1.25^b^	1.66 ^a^	1.58 ^a^	0.08	<0.001	<0.001	0.063
SFA	40.90 ^a^	39.37 ^b^	38.78 ^b^	38.79 ^b^	0.38	0.003	0.001	0.058
USFA	57.67 ^b^	58.5 ^ab^	60.05 ^a^	58.96 ^ab^	0.38	0.003	0.001	0.058
MUFA	30.05	30.47	30.85	30.73	0.25	0.144	0.043	0.287
PUFA	27.62 ^b^	28.03 ^ab^	29.19 ^a^	28.23 ^ab^	0.32	0.024	0.007	0.143
USFA: SFA	1.41 ^b^	1.49 ^ab^	1.55 ^a^	1.52 ^a^	0.03	0.005	0.001	0.069

Values in the same row with no letter or the same letter superscripts mean no significant difference (*p* > 0.05), while with different letter superscripts mean significant difference (*p* < 0.05).T1: the broilers were fed a basal diet; T2: the broilers were fed a basal diet supplemented with AGE at 0.2%; T3: the broilers were fed a basal diet supplemented with AGE at 0.4%; T4: the broilers were fed a basal diet supplemented with AGE at 0.6%.SFA, saturated fatty acid; USFA, unsaturated fatty acids; MUFA, monounsaturated fatty acids; PUFA, polyunsaturated fatty acids; USFA: SFA, the ratio of unsaturated fatty acids to saturated fatty acid.

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
