# Peer review of "Effects of Vine Tea Extract on Meat Quality, Gut Microbiota and Metabolome of Wenchang Broiler"

_animals, 2022, doi:10.3390/ani12131661_

Round 1
Reviewer 1 Report
I read the manuscript " Effects of Ampelopsis grossedentata extract on meat quality, gut 2 microbiota and metabolome of Wenchang broiler ". I believe that the manuscript's contents are very interesting and fit the scope of the journal. Overall, from hypothesis to conclusion, the study is compact and well presented.
The manuscript is interesting; I have only a few comments before publication:
- Add research background problem and their possible solution in the introduction section.
- How did the authors select the dose of supplementation, please explain?
- How the author’s select experimental period is any reference or preliminary study?.
- Animal ethics approval code is missing, is this study approved by ethics committee?
Author Response
Question: Add research background problem and their possible solution in the introduction section.
Answer: We add some express of research background and their possible solution in the introduction section. Seen Line 46-53.
Question: How did the authors select the dose of supplementation, please explain.
Answer: We selected the dose of supplementation refer to the dosage recommended by the product manufacturer. The recommended dosage is 0.2% to 0.6%. So, we set the three doses of supplementation in the present study (0.2%, 0.4% and 0.6%).
Question: How the author’s select experimental period is any reference or preliminary study?
Answer: Local chicken farmers generally fatten Wenchang broilers from 70-day-old to 120-day-old, and then sale to market. The purpose of this study is to investigate the effect of adding Ampelopsis grossedentata extract (AGE) on the quality and flavor of Wenchang broilers during the fattening period.
Question: Animal ethics approval code is missing, is this study approved by ethics committee?
Answer: This study approved by the Institutional Animal Care and Use Committee of the Chinese Academy of Tropical Agricultural Sciences. The approval number is CATAS-20201015-1. We added the approval code in the manuscript.
Reviewer 2 Report
Manuscript animals-1751172, entitled “Effects of Ampelopsis grossedentata extract on meat quality, gut microbiota and metabolome of Wenchang broiler”
Recommendation: The above paper is not suitable for publication in its present form.
The article provides useful information about the effects of of Ampelopsis grossedentata extract on meat quality, gut microbiota and metabolome in Wenchang broilers. Although, the experiment was in general appropriately designed and implemented, there are some points that should be corrected or clarified.
General comments
· Please add a paragraph (L44-51) referring to the fact that the therapeutic use of antibiotics is banned in EU since 2006 and that other top country producers are emerged to do the same.
· Please be specific and provide scientific names of muscles (for example, Pectoralis major, Biceps femoris etc) (L93-102)
· Please provide experimental duration in Material and Methods (L80-82). The day in L93 refers to experimental period or age of broilers?
· L144-150: Preparation of meat samples for shear force? How was the standardization of chromameter, pHmeter and dynamometer accomplished?
· Please explain how Inosine monphosphate analysis is related with meat quality
· L165-170: Please provide more GC-MS details
· Please try to be specific:
L236: at 0.2 and 0.4%. What about the quadratic effect
L250: at 0.2 and 0.4 %.
L262: Please add “…than the T1 group (also T2 in breast), while…” and delete L266-267.
L263: Not for T4 in breast
L286: Where are these data shown?
L370-371: “…of AGE quadratically decreased serum TCHO, and linearly reduced TG and LDL-C, and linearly increased serum HDL-C concentrations…”
L388-389: T3 and T4, not all three groups. Pleas check values of drip loss in Table 3
L399: T3 and T4
· Please explain why T2 generally did not show significant improvements
Minor points
L15: Please delete “of Wenchang broilers”
L34: “control” instead of “T1 group”
L34-37: The same as in simple summary. Please rephrase
L55: “…with the high levels…”
L79: “allocated” instead of “divided”
L82: Please delete “respectively”
L83-84: “Feed and water were ad libitum offered to broilers.”
L85: “…AGE mainly included…”
L96: “…for subsequent analysis.”
L101: “…were also collected…”
L109, 118: “of approximately” instead of “about”
L130: “…as previously described [23].”
L180: Please rephrase
L205, 224: Please delete “respectively”
L228: “are” instead of “were”
L234-235: Please delete “was linearly reduced”
L246: “The effects of AGE dietary supplementation on meat…”
L262: “…(SFA) (especially C14:0 and C16:0) levels…”
L263: “…(PUFA) (mainly C22:6n3) content…”
L265: “C16:1n7” and “C20:1n9”
L300: “To investigate the different metabolic pathway in the…”
L301: “observed” instead of “seen”
L302-303: Please rephrase
L308: “observed in” instead of “seen to”
L309-310: “It suggested that AGE supplementation at 0.4% significantly downregulated…”
L336: “molecule”
L381: “AS it is known, the activation…”
L423: “In addition, the present study…”
L431: “improve”
Author Response
Question: Please add a paragraph (L44-51) referring to the fact that the therapeutic use of antibiotics is banned in EU since 2006 and that other top country producers are emerged to do the same.
Answer: We added the regulation policies of antibiotics use in EU and China. Seen Line 47-49.
Question: Please be specific and provide scientific names of muscles (for example, Pectoralis major, Biceps femoris etc) (L93-102)
Answer: We added the scientific names of leg and breast muscles. Seen Line 101.
Question: Please provide experimental duration in Material and Methods (L80-82). The day in L93 refers to experimental period or age of broilers?
Answer: The feeding trial lasted 54 days (from the 70-day-old to 124-day-old of broilers). The day in L93 refers to experimental period. We added the experimental duration.
Question: L144-150: Preparation of meat samples for shear force? How was the standardization of chromameter, pH meter and dynamometer accomplished?
Answer: Preparation of meat samples and measurement methods we referred to Jiang et al. (2011). We have added a more detailed operation method. Seen Line 149-169.
Question: Please explain how Inosine monphosphate analysis is related with meat quality.
Answer: IMP is an endogenous compound in muscle. It is well known to affect meat flavor and IMP contributes to the umami taste, alone or conjugated with monosodium glutamate for synergistic effects. Therefore, increasing IMP concentration in meat may improve the sensory quality of meat (Jung et al., 2013). We added clarification in the discussion section. Seen Line 418-421.
Jung, S.; Bae, Y.S.; Kim, H.J.; Jayasena, D.D.; Lee, J.H.; Park, H.B.; Heo, K.N.; Jo, C. Carnosine, anserine, creatine, and inosine 5'-monophosphate contents in breast and thigh meats from 5 lines of Korean native chicken. Poult. Sci. 2013, 92, 3275-3282.
Question: L165-170: Please provide more GC-MS details
Answer: We added more GC-Ms details. Seen L183-198.
Question:
L236: at 0.2 and 0.4%.
L250: at 0.2 and 0.4 %.
Answer: Corrected.
.
Question: L262: Please add “…than the T1 group (also T2 in breast), while…” and delete L266-267.
Answer: Corrected.
Question: L263: Not for T4 in breast
Answer: OK, we corrected.
Question: L286: Where are these data shown?
Answer: Alpha diversity indexes between T1 and T3 group are not significantly different. The original statement was not rigorous. So, we deleted the original express.
Question: L370-371: “…of AGE quadratically decreased serum TCHO, and linearly reduced TG and LDL-C, and linearly increased serum HDL-C concentrations…”
Answer: Revised.
Question: L388-389: T3 and T4, not all three groups. Please check values of drip loss in Table 3
Answer: Corrected.
Question: L399: T3 and T4. Please explain why T2 generally did not show significant improvements.
Answer: Corrected. Although the content of total PUFA was increased in the T2 group compared with the T1 group, no significant difference was seen between the T1 and T2 groups. This may be related to the fact that T2 group was characterized by the lowest flavonoids content among the three AGE groups. Seen Line 427-430.
Question:
L15: Please delete “of Wenchang broilers”
L34: “control” instead of “T1 group”
L34-37: The same as in simple summary. Please rephrase
L55: “…with the high levels…”
L79: “allocated” instead of “divided”
L82: Please delete “respectively”
L83-84: “Feed and water were ad libitum offered to broilers.”
L85: “…AGE mainly included…”
L96: “…for subsequent analysis.”
L101: “…were also collected…”
L109, 118: “of approximately” instead of “about”
L130: “…as previously described [23].”
L180: Please rephrase
L205, 224: Please delete “respectively”
L228: “are” instead of “were”
L234-235: Please delete “was linearly reduced”
L246: “The effects of AGE dietary supplementation on meat…”
L262: “…(SFA) (especially C14:0 and C16:0) levels…”
L263: “…(PUFA) (mainly C22:6n3) content…”
L265: “C16:1n7” and “C20:1n9”
L300: “To investigate the different metabolic pathway in the…”
L301: “observed” instead of “seen”
L302-303: Please rephrase
L308: “observed in” instead of “seen to”
L309-310: “It suggested that AGE supplementation at 0.4% significantly downregulated…”
L336: “molecule”
L381: “AS it is known, the activation…”
L423: “In addition, the present study…”
L431: “improve”
Answer: The parts mentioned above have been revised, seen the red marked part in the paper.
Round 2
Reviewer 2 Report
Authors made the majority of the necessary amendments. Some minor points should be corrected before the acceptance of their submission:
L411-413: NS differences. Please rephrase
L430: "observed" instead of "seen"
L426: Only T3
Author Response
Question: L411-413: NS differences. Please rephrase
Answer: We rephrased it as “The T3 and T4 groups had higher a* value both in thigh and breast muscles than the T1 group. Additionally, the dietary supplementation of AGE linearly decreased the shear force and both in thigh and breast muscles than the T1 group”.
Question: L430: “observed” instead of “seen”
Answer: Corrected.
Question: L426, only T3.
Answer: Revised.